# Optimization of Cellular Transduction by the HIV-Based Pseudovirus Platform with Pan-Coronavirus Spike Proteins

**DOI:** 10.3390/v16091492

**Published:** 2024-09-20

**Authors:** Syamala Rani Thimmiraju, Maria Jose Villar, Jason T. Kimata, Ulrich Strych, Maria Elena Bottazzi, Peter J. Hotez, Jeroen Pollet

**Affiliations:** 1Texas Children’s Hospital Center for Vaccine Development, Houston, TX 77030, USA; syamala.thimmiraju@bcm.edu (S.R.T.); mariajose.villarmondragon@bcm.edu (M.J.V.); strych@bcm.edu (U.S.); bottazzi@bcm.edu (M.E.B.); hotez@bcm.edu (P.J.H.); 2Department of Pediatrics, National School of Tropical Medicine, Baylor College of Medicine, Houston, TX 77030, USA; 3Department of Molecular Virology and Microbiology, Baylor College of Medicine, Houston, TX 77030, USA; jkimata@bcm.edu

**Keywords:** coronaviruses, lentiviral pseudoviruses, transduction efficiency, neutralizing antibodies, spinoculation, polybrene

## Abstract

Over the past three years, new SARS-CoV-2 variants have continuously emerged, evolving to a point where an immune response against the original vaccine no longer provided optimal protection against these new strains. During this time, high-throughput neutralization assays based on pseudoviruses have become a valuable tool for assessing the efficacy of new vaccines, screening updated vaccine candidates against emerging variants, and testing the efficacy of new therapeutics such as monoclonal antibodies. Lentiviral vectors derived from HIV-1 are popular for developing pseudo and chimeric viruses due to their ease of use, stability, and long-term transgene expression. However, the HIV-based platform has lower transduction rates for pseudotyping coronavirus spike proteins than other pseudovirus platforms, necessitating more optimized methods. As the SARS-CoV-2 virus evolved, we produced over 18 variants of the spike protein for pseudotyping with an HIV-based vector, optimizing experimental parameters for their production and transduction. In this article, we present key parameters that were assessed to improve such technology, including (a) the timing and method of collection of pseudovirus supernatant; (b) the timing of host cell transduction; (c) cell culture media replenishment after pseudovirus adsorption; and (d) the centrifugation (spinoculation) parameters of the host cell+ pseudovirus mix, towards improved transduction. Additionally, we found that, for some pseudoviruses, the addition of a cationic polymer (polybrene) to the culture medium improved the transduction process. These findings were applicable across variant spike pseudoviruses that include not only SARS-CoV-2 variants, but also SARS, MERS, Alpha Coronavirus (NL-63), and bat-like coronaviruses. In summary, we present improvements in transduction efficiency, which can broaden the dynamic range of the pseudovirus titration and neutralization assays.

## 1. Introduction

SARS-CoV-2 is a class 3 risk group beta-coronavirus [1] responsible for the COVID-19 pandemic. Like other epidemic pathogens, coronaviruses elicit neutralizing antibodies in the host in response to infection or vaccination. Studies have suggested such neutralizing antibodies represent potential correlates of immunological protection against COVID-19 disease [2,3,4,5,6]. Due to the dominant immunogenic nature of the spike or “S” glycoprotein of coronaviruses, it is used as a component of multiple vaccines [7,8]. Assays that can measure the serological response to the spike glycoprotein are crucial for vaccine efficacy studies. After an initial wave of infections by the SARS-CoV-2 ancestral strain, the virus rapidly evolved, producing variant strains with multiple mutations. Many of these were proven to be less susceptible to neutralization by convalescent or post-vaccination sera against original strain, increasing disease transmission and severity [9,10]. The quantification of neutralizing antibodies against live viral versions of these emerging variants is slow, tedious, and requires Bio Safety Level 3 (BSL-3) facilities. Therefore, in vitro assays under Bio Safety Level 2 (BSL-2) conditions helped screen vaccine efficacies against these new mutant variants. Currently, various forms of Enzyme-Linked Immunosorbent Assays (ELISAs) or their derivatives are being used for such purposes [11,12,13]. Pseudovirus neutralization assays in vitro provide an attractive platform by mimicking the live virus infection (transduction), allowing the characterization of emerging virus variants and the evaluation of novel therapeutics such as monoclonal antibodies [14,15].

Improvements in pseudovirus transduction efficiencies in vitro aid the accurate titration of pseudovirus particles packaged in host cells after transient co-transfection. The successful optimization of experimental parameters towards the production of high pseudovirus titers and establishing robust neutralization assays for screening vaccines and therapeutics would benefit from improved transduction protocols. Lentiviruses, like human immunodeficiency virus (HIV)-based platforms, offer suitable systems to generate pseudoviruses of enveloped viral pathogens [16]. Studies showed that the neutralizing titers of antibodies and sera measured by pseudoviruses were highly correlated with those measured by live viruses [17,18]. SARS-Cov-2 spike-pseudovirus neutralization correlates with the neutralization of infectious SARS-CoV-2 [19] and is important for its continued use for vaccine development and for phenotyping escape mutations in the spike. Using an HIV-based platform, we have generated luciferase-expressing, replication-incompetent coronavirus pseudoviruses mimicking the spike proteins of SARS-CoV, MERS-CoV, various bat-like coronaviruses, and SARS-CoV-2 and its variants. However, the HIV-based platform exhibits lower transduction rates for pseudotyped coronaviruses compared to other pseudovirus platforms like vesicular stomatitis virus (VSV) [20], highlighting the need for more optimized methods. Therefore, we have optimized in vitro pseudoviral packaging parameters and transduction efficiencies. We also compare the infectivity of pseudotyped lentiviral particles representing the spike gene of ten different coronavirus variants to assess the efficacy and applicability of our assays across multiple variants of concern.

Previously described viral titration and neutralization assays [21,22,23] were used as guidance to determine which assay parameters to optimize towards high pseudoviral titers, low assay variability, and maximum dynamic range. The optimized parameters (Table 1) include (1) the timing of pseudovirus supernatant collection, (2) the method of pseudovirus supernatant collection, (3) the timing of transduction after host–cell seeding, (4) the replenishment of media after pseudovirus adsorption, (5) the centrifugation (spinoculation treatment) of the host cells immediately after pseudovirus addition, and (6) the effect of polybrene (Hexadimethrine bromide) to normalize the charge repulsion between viral and host cell membranes to better facilitate in vitro transduction.

## 2. Materials and Methods

### 2.1. Cell Lines Used for Pseudovirus Production and Transduction

For co-transfection (with packaging/reporter and spike plasmids) and production of pseudoviruses, HEK-293T cells (ATCC CRL-3216) were used [24]. For titrating SARS- and MERS-pseudovirus particles and for evaluating the neutralization efficacy of sera samples, HEK-ACE2 and HeLa-DPP4 [25] cells were used, respectively. HEK-ACE2 was obtained through BEI Resources, an NIH-supported program managed by the ATCC: Human Embryonic Kidney Cells (HEK-293T) Expressing Human Angiotensin-Converting Enzyme 2 (HEK-293T-hACE2 Cell Line, NR-52511). To generate a human DPP4 (CD26)-expressing target cell line (Hela-DPP4) for infection with MERS spike pseudovirions, we transduced HeLa cells with a DPP4 encoding lentiviral vector, pLEX307-DPP4-puro (pLEX307-DPP4-puro was a gift from Alejandro Chavez and Sho Iketani (Addgene (Watertown, MA, USA) plasmid #158451; http://n2t.net/addgene:158451 (6 July 2022); RRID:Addgene_158451)) and selected for vector-expressing cells using 1 µg/mL puromycin (MilleporeSigma (Burlington, MA, USA), P4512). Puromycin-resistant cells were analyzed for DPP4 (CD26) surface expression by flow cytometry on an Attune Acoustic Focusing Cytometer using a mouse anti-human CD26-PE antibody (clone BA5b, Biolegend (San Diego, CA, USA), 302705) and found to be approximately 90% DPP4-positive. Cells were live-cell-sorted on a BD FACSDiscover S8 Cell Sorter for high DPP4 expression, re-analyzed, and shown to be greater than 99% positive for DPP4.

### 2.2. Plasmid Constructs Used to Produce Pseudoviruses

The plasmids used for the pseudovirus production were the luciferase-encoding reporter plasmid pNL4-3.Luc-R-E-, the Gag/Pol-encoding packaging construct pΔ8.9, and codon-optimized spike protein genes expressed from pcDNA3.1 mammalian expression vectors [26]. pcDNA3.1-XBB1.5 was used to express the SARS-CoV-2 variant of concern XBB1.5 [27]. Plasmids that express the coronavirus spike proteins for MERS, NL63, WIV1, and SARS-CoV-1 were obtained from Addgene. pCDNA3.3_MERS_D12 was a gift from David Nemazee (Addgene plasmid #170448; http://n2t.net/addgene:170448 (6 July 2022); RRID:Addgene_170448) [28] pcDNA3.3-NL63-D14 was a gift from David Nemazee (Addgene plasmid #172666; http://n2t.net/addgene:172666 (6 July 2022); RRID:Addgene_172666) [29] pTwist-SARS-CoV Δ18 was a gift from Alejandro Balazs (Addgene plasmid #169465; http://n2t.net/addgene:169465 (7 October 2021); RRID:Addgene_169465) [30]. To express RSsHC014 and RatG13 spikes, we codon-optimized sequences of RSsHC014 and RatG13 spike genes. The last 3′ 19 codons were removed to improve incorporation into lentiviral pseudovirions, and a flag-tag was added for detection by Western blot. Genscript (Piscataway, NJ, USA) synthesized and inserted the genes into the pcDNA3.1(+) vector (pcDNA3.1-CoV-RaTG13del19 and pcDNA3.1-CoVRsSHC014del19). In total, 18 coronavirus-variant pseudoviruses were generated and tested: SARS1, D614G, alpha, beta, delta, lambda, BA1, BA2, BA 4, BA 6, BA 2.75, BQ 1.1, XBB 1.5, XBB 1.16, EG 5, JN 1, W1V1, W1V16, RATG13, RsSHCO14, MERS-CoV, and NL-63 (Table 2).

### 2.3. Production and Titration of Pseudoviruses

Pseudovirus particles were produced (Figure 1) using the HEK-293T cell line, as described earlier [19]. Briefly, cells were cultured in low-glucose Dulbecco’s modified Eagle medium (DMEM, Life Technologies) supplemented with 10% fetal bovine serum (FBS), penicillin (50 U/mL), and streptomycin (50 µg/mL) at 37 °C in a humidified atmosphere with 5% CO_2_. Replication-deficient HIV-based pseudoviruses bearing coronavirus spike proteins were generated as described [27] previously. HEK-293T cells were transfected with expression plasmids encoding HIV Gag/Pol, individual coronavirus spike genes, and firefly *luciferase* as a reporter. For optimizing the time point of pseudovirus collection post-transfection, supernatants were harvested either at 48 h or 72 h. Similarly, to compare the method of pseudovirus harvest, the pseudovirus-containing supernatants were either filtered (0.45 μm pore size; Merck Millipore, Burlington, MA, USA) or centrifuged (1200× *g* for 8 min) and stored in 2 mL aliquots at −80 °C until further use. All frozen stocks were used only once after thawing to avoid inconsistencies resulting from repeated freezing–thawing cycles.

For the titration of pseudovirus particles (Figure 1), Poly-D-lysine-coated 96-well flat bottom plates were seeded either 4–6 h or 16–18 h before the planned transduction with either HEK-ACE2 cells or HeLa-DPP4 cells (1.5 × 10^4^/well) in 100 µL DMEM. Prior to the process of transduction, a pseudovirus dilution plate was prepared. For this, an aliquot of pseudovirus stock was diluted twofold initially with DMEM, and 200 µL/well of this was added into column #2 of a 96-well plate and further serial-diluted (2-fold) 8 more times, making the dilution range from 4-fold to 512-fold, and dispensed horizontally, starting from the left columns to the right (column #3–9). Each dilution point was made in six replicate wells vertically from the top to bottom of each column (B–G) of a 96-well culture plate. The column #10 served as the “no pseudovirus control” column with 100 µL of DMEM only. Thus, by the end of this process, each well had a volume of 100 µL of a specific dilution of pseudovirus. For host cell transduction, 50 µL from each well of this pseudovirus dilution plate was added to the exact corresponding wells of the pre-seeded host cell plate, and incubated at 37 °C and 5% CO_2_ for 48 h.

After 48 h of incubation, the cell culture supernatant was aspirated, and cells were washed with 100 µL of PBS followed by cell lysis with 100 µL Promega Glo Lysis buffer (Cat. #E2661) for 15 min at RT. Finally, 50 µL of the lysate was added to 50 µL luciferase substrate (Promega Luciferase Assay System, Cat. #E1500). The amount of luciferase is directly proportional to the number of transduced cells and quantified as luminescence (Relative Luminescence Units, RLU) using a Luminometer (BioSynergy-H4). The positive well was determined as tenfold RLU values higher than the background (no pseudovirus control) value. The concentration of pseudovirus particles and the 50% tissue culture infectious dose (TCID50) were calculated using the Reed–Muench method, as described previously [16,20].

### 2.4. Use of Polybrene to Facilitate Transduction

To study the effect of a cationic polymer on pseudovirus transduction efficiency, we added polybrene (Hexadimethrine bromide, Millipore Sigma, (Burlington, MA, USA), P4512), (Cat. #H9268-5G), at a concentration of 4 µg/mL of DMEM cell culture medium. However, polybrene was not used during the initial optimization experiments for other parameters (pseudovirus collection time, collection method, early vs. late infection, and spinoculation experiments). The polybrene effect was only assessed with the new protocol in specific experiments to compare transduction frequencies using original and new protocols across ten coronavirus spikes.

### 2.5. Neutralization of Pseudoviruses Using Vaccinated/Convalescent Sera

Neutralization was scored as the reduction in luciferase gene expression as described previously [19]. Four to six hours before the neutralization assay, target host cells (293T-ACE2 or DPP4 cells) were seeded in Poly-D-lysine-coated 96-well flat bottom plates (1.5 × 10^4^/well in 100 µL DMEM). Sera samples were always heat-inactivated for 30 min. at 56 °C before their use in the neutralization assays. Two hours before the transduction of the cells with pseudovirus, a “sera dilution plate” was prepared using a U-bottomed, tissue-culture-treated 96-well transparent plate with a 10-fold initial dilution of heat-inactivated test sera samples (10 µL of sera and 90 µL of DMEM with 10%FBS), and further 4-fold serial dilution (25 µL from top well added to 75 µL of DMEM in the immediate bottom well) to make a total of 8 dilutions per sample. Next, a “pseudovirus plate” (a U-bottomed, tissue-culture-treated 96-well transparent plate was prepared with the same input volume of 30 µL in each experimental well (corresponding to ~3–5 × 10^6^ RLUs/mL)). Therefore, all of the experimental wells in the 96-well pseudovirus plate had 30 µL of diluted pseudovirus, except the “no pseudovirus” control wells that had 60 µL of DMEM each.

For the pseudovirus neutralization step, 30 µL of serial-diluted test sera from each well of the “dilution plate” were added to the corresponding well in the “pseudovirus plate” (with 30 µL pseudovirus), making a total final volume of 60 µL in each experimental well. This addition of an equal volume of pseudovirus also made the initial dilution of serum sample 20-fold instead of 10-fold. Thus, the 8 final serial dilutions of sera ranged from 20 to 327,680. These pseudovirus–sera mixtures were incubated for 1 h at 37 °C and 5% CO_2_ to facilitate the neutralization of the pseudoviruses by antibodies, if any, from the test sera (Figure 1). After 1 h incubation, 50 µL of this sera–pseudovirus mix was added to the target cells (293T-hACE2 or DPP4) that were pre-seeded earlier in 96-well plates. In the case of the experiments that included a spinoculation step (new protocol), these host cell plates with sera–pseudovirus mixture added on top were then subjected to a centrifugation step for 60 min at 200× *g*. After spinoculation, the cell plates were incubated for 48 h in a 5% CO_2_ environment at 37 °C. Post 16–18 h incubation, the media with the remaining pseudovirus–sera mix was removed, and the cells were replenished with fresh DMEM media and incubation continued (for a total of 48 h after spinoculation).

Following the 48 h incubation, the cell culture supernatant was aspirated, and the cells were washed with 100 µL of PBS followed by cell lysis with 100 µL Promega Glo Lysis buffer (Cat. #E2661) for 15 min at RT. Finally, 50 µL of the lysate was added to 50 µL luciferase substrate (Promega Luciferase Assay System, Cat. #E1500). The amount of luciferase was quantified by luminescence (Relative Luminescence Units (RLU)) using a Luminometer (BioSynergy-H4). The conditions were tested in duplicate wells on each plate, and a virus control (VC = no sera) and cell control (CC = no pseudovirus) were included on every plate in 8 wells each to represent the values of 0% and 100% neutralization, respectively, for data normalization purposes. Based on these results, the IC50 of each sample was calculated according to the method described by Nie et al. [20,31]. The 50% inhibitory dilution was determined as the sera dilution at which the RLUs were decreased by 50% compared with viral control wells (virus + cells), after subtracting background RLUs from cell-only controls. The percentage of inhibition of infection for each dilution of the sample was calculated according to the RLU values as follows:

% inhibition = [1 − (average RLU of sample − average RLU of Viral Controls)/(Average RLU of Viral Controls − average RLU of Cell Controls)] × 100%.


## 3. Results

### 3.1. Effect of Incubation Time and Method of Collection of Pseudovirus Supernatant on Transduction Efficiency (Pseudovirus Titer)

After co-transfecting host cells with an HIV packaging plasmid, a luciferase reporter plasmid, and one of the four variant spike plasmids (D614G, Beta, Delta, and Omicron BA.4), the cultures were incubated for either 48 h or 72 h before collecting the pseudovirus supernatant to study the transduction efficiency (as reflected by luminescence = RLU numbers). While the SARS-CoV-2 variants differed in their RLU values for all four variants studied, collecting pseudovirus supernatant at 48 h significantly improved transduction (Figure 2A), yielding higher titers of infectious pseudoviruses. Additionally, two collection methods were compared to collect the secreted pseudoviral particles. The culture supernatants were either filtered through a 0.45 μm filter or centrifuged for 8 min at 1200× *g* to remove debris and were aliquoted before being stored at −80 °C for future use. For all four variants studied, centrifugation treatment without filtering the supernatant gave equal or better RLU numbers than filtering the supernatant (Figure 2B). This improvement was especially prominent for the Beta variant with a fold difference of 9.13 (Appendix A). Therefore, for our improved protocol, we collected pseudoviruses after 48 h by centrifugation.

### 3.2. Effect of Host Cell Transduction Timing after Cell Seeding on Transduction Efficiency

For transducing the cells with variant pseudoviruses, host cells were seeded either 16–18 h prior (termed as late transduction), or 4–6 h (termed as early transduction) before transduction. For the four SARS-CoV-2 variants evaluated (D614G, Beta, Delta, and Omicron BA.4), early transduction at the 4–6 h time point resulted in a marked improvement in RLU values (Figure 3), ranging from a 10-fold increase for the Beta variant to a 54-fold increase for the Delta variant (Appendix A). Therefore, in our new protocol, we transduced the host cells at the 6–8 h time point.

### 3.3. Effect of Changing the Media 16–18 h after Transduction/Viral Adsorption

For both variants tested (D614G and BA4), removing the pseudovirus–sera mixture 16–18 h after transduction and replenishing the host cells with DMEM media increased the RLU numbers (Figure 4). Therefore, in our new protocol, we replenished the cell culture media after the initial adsorption of the pseudovirus after 16–18 h.

### 3.4. Optimization of Spinoculation Parameters

Spinoculation (centrifugation) was successfully used in previous studies [32,33] to increase pseudovirus transduction rates. The spinoculation process increases the radial centrifugal force on the viral particles in suspension, making them more accessible to target cells. Therefore, a combination of two centrifugation speeds (200× *g* and 400× *g*) and three-time points (5, 30, and 60 min) were evaluated at room temperature to study whether spinoculation treatment affected transduction. For both the BA 4 and BQ 1.1 variants studied, spinoculation for 60 min enhanced the transduction numbers significantly (Figure 5A). However, changing the centrifugation speed did not significantly improve the RLUs (Figure 5A). Thus, including a spinoculation step of 200× *g* centrifugation for 60 min. in our new protocol improved the transduction efficiency (Figure 5B) when tested with additional coronavirus variants (SARS1, D614G, MERS-CoV, Alpha-CoV, and bat pseudoviruses RsSHC014 and RaTG13).

### 3.5. Comparison of Original and New Protocols for Titration of Pseudovirus Particles

The pseudovirus production and transduction efficiencies in the titration assays were compared by conducting experiments according to the previously published protocol [34] (hereafter, called the original protocol) and incorporating optimized parameters from our current study (called the new protocol) for four spike variants (D614G, Beta, Delta, and Omicron BA.4). Both protocols stayed true to robust linear regression values of dose–response, indicating the linearity between increased dilutions of pseudovirus and decreased RLU numbers (R > 0.97). However, for all four variants studied, the RLU values were enhanced at least by a log-fold (Figure 6) using the new protocol, thereby improving our titration assay with five dilution points in the linear range (8-fold to 128-fold dilution), compared to the three dilution points in the linear range (4-fold to 16-fold dilution) in the original protocol (Figure 6). Therefore, using this new protocol, we produced and titrated more than 18 variant pseudoviruses, spanning a wide range of infectious particles (TCID50/mL) and luminescence (RLU/mL), as shown in Table 2.

### 3.6. Comparison of Transduction Efficiency of Coronavirus Pseudoviruses Using Original and New Protocols across 10 Spike Variants

After improving transduction numbers by modifying some experimental parameters of the original protocol as described above (a minimum of two variant pseudoviruses were studied for evaluating each parameter), we compared the original protocol with our new protocol across ten coronavirus variant pseudoviruses. Additionally, polybrene has been widely used in neutralization and infectivity assays examining HIV and pseudo-typed lentiviruses coated with the SARS-CoV-2 S protein [19,21,35,36]. Polybrene, being a cationic polymer, can enhance in vitro HIV-1 infection by reducing the electrostatic repulsion between virions and sialic acid on the cell surface. Therefore, we also assessed polybrene’s role in improving the RLU numbers for these 10 variants when used with the new protocol. Our results demonstrate an improvement in transduction numbers by at least one log-fold with the new protocol compared to the original protocol (Figure 7). Additionally, for at least 9 of the 10 variants tested, polybrene’s addition to the new protocol resulted in equal or better RLU numbers (Figure 7). To test whether such improvements in our assay’s dynamic range (increased RLUs) resulted in decreased assay variability, we employed coefficient of variation (CV%) analyses to assess variability within replicates in each of the three protocols (Figure 8). We observed a CV% range of 13–66 in the original protocol with an average CV% of 21.5 across 10 spike-variant pseudoviruses, and ranging from 5–16 in the new protocol with an average of 8.5 across 10 spike-variant pseudoviruses. Similarly, the addition of polybrene to our new protocol resulted in a CV% range of 4–13 with an average CV% of 7.1 across 10 spike variant pseudoviruses.

Additionally, to evaluate whether the positive effect of polybrene in transduction could successfully be employed in our neutralization assays, we performed virus neutralization experiments in the presence or absence of polybrene, using four pseudovirus variants (D614G, Beta, Delta, and Omicron). Our results (Appendix A) indicated that polybrene, while facilitating an efficient transduction process, did not interfere with neutralization activity between pseudovirus and the sera. Except for the Delta variant, polybrene improved transduction numbers for the other variants, thereby improving the dynamic range of the neutralization assay.

## 4. Discussion

The HIV-1-based pseudovirus packaging platform is a popular model for developing pseudoviruses due to their ease of use and stability. However, studies show that in comparison to the avesicular stomatitis virus (VSV)-based pseudotype packaging system, where the backbone comes from the VSV virus (the G gene is replaced with a reporter gene, and an envelope protein from SARS-CoV-2 is incorporated), the HIV-based platform has lower transduction rates (RLU values) [20,37,38].

The results from our current study using an HIV-based pseudovirus platform demonstrate that there is a significant improvement in the recovery of high-titer spike pseudoviruses with our new protocol, and that treatments like spinoculation, early host cell transduction, and polybrene addition enhance transduction efficiencies across a wide range of variant pseudoviruses. The improvement of RLU numbers, at least by a log-fold, helps the assay to rise above noise levels for the titration of pseudovirus particles and enables us to better normalize a broad range of spike variant pseudoviruses to arrive closer to their actual TCID50s. This way, we are more likely to add equal amounts of infectious virus for the different coronavirus spike variants for neutralization assays. Achieving more linear points in the pseudovirus titration curves enables us to use a lower input dose of pseudovirus without compromising the transduction efficiencies. Effective transduction at such lower multiplicities of infection (MOIs) allows the assays to be performed with fewer viral particles, which is immensely helpful when viral stock is limited (emerging pathogens) or difficult to produce.

These improvements also translate into better neutralization assays because having a low-titer infectious virus makes it difficult to add enough virus–sera combinations for the neutralization assays. Concentrating the pseudoviral supernatant may not be a successful option across all spike variants due to the need for scaled-up production volume to do so, and due to the possible loss of viral particles during the centrifugation process. By improving transduction efficiencies, we can elevate the assay’s dynamic range to rise above the background noise, leading to the better calculation of neutralization titers and decreasing the chance of false-negative rates. While our new protocol helped to enhance consistency in the replicates, the addition of polybrene may be particularly useful for spike variants that package very low titers, as observed in NL63, RsSHC014, and RaTG13.

The improved protocol is expected to be beneficial for assessing pan-coronavirus entry inhibitors, evaluating the neutralizing antibody activity of vaccinated/convalescent antisera, and “anti-S” protein monoclonal antibodies and therapeutics. With pan-coronavirus universal vaccine development progressing at a great pace, the optimization of transduction efficiencies across pan-coronavirus pseudoviruses is valuable for screening the vaccine candidates and testing the efficacy of next-generation pan-coronavirus vaccines through pseudovirus neutralization assays.

## Figures and Tables

**Figure 1 viruses-16-01492-f001:**
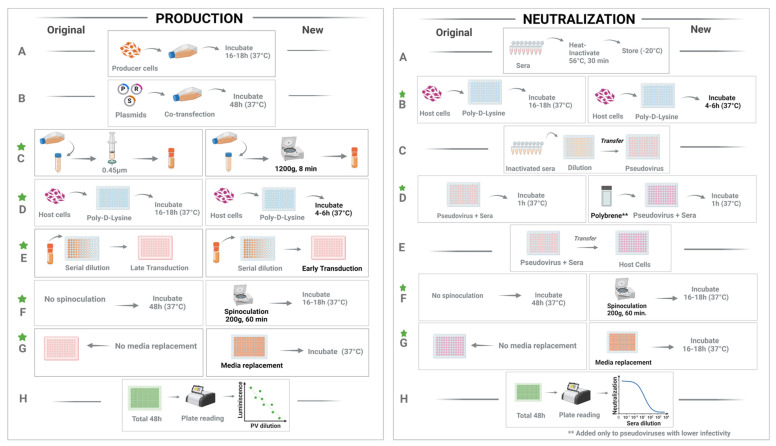
Important steps (**A**–**H**) in pseudovirus production (*left*) and neutralization (*right*). (**A**) producer cell incubation (*left*)/sera heat inactivation (*right*) (**B**) co-transfection (*left*)/incubation (*right*) (**C**) pseudovirus supernatant collection (*left*)/neutralization (*right*) (**D**) Host cell incubation (**E**) Host cell Transduction (**F**) Spinoculation and incubation (**G**) Media replenishment (**H**) Luminescence reading. Within each panel, the left side depicts the original protocol, and the new protocol is shown on the right. In step B of the production panel, P = packaging plasmid, R = reporter plasmid, and S = spike envelop plasmid. The steps where experimental factors were optimized are indicated by green stars.

**Figure 2 viruses-16-01492-f002:**
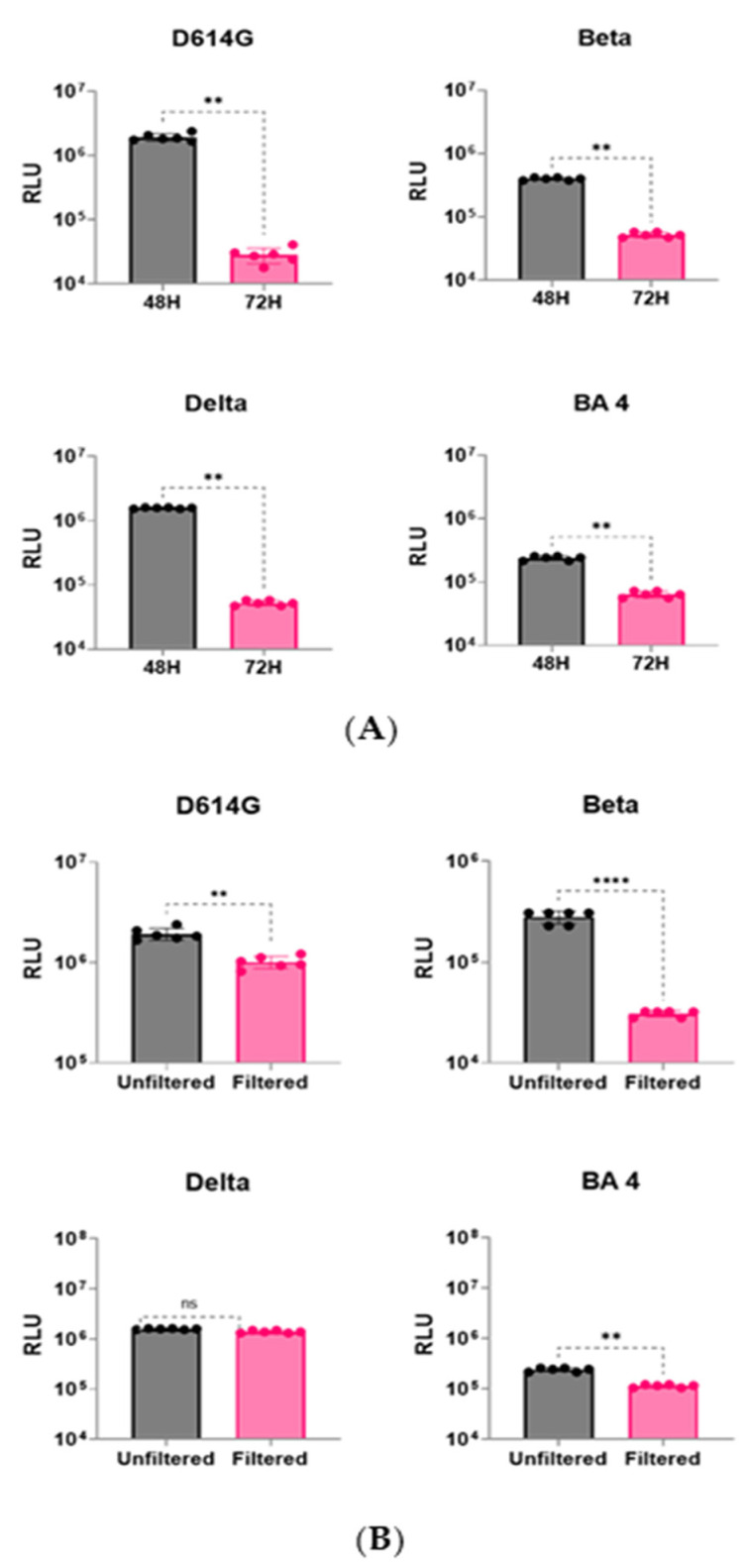
(**A**) Timing of pseudovirus supernatant collection. The bar graph compares the timing of pseudovirus collection on transduction efficiency, reflected by luminescence (RLU). Data were analyzed for significance (*p* < 0.05) using a two-tailed Mann–Whitney test. For all four variants studied, collecting pseudovirus supernatant at 48 h significantly (** represents *p* < 0.01) enhanced transduction numbers over collection at 72 h. (**B**) Method of supernatant collection. For all four variants studied, unfiltered supernatant showed equal or better RLU numbers than filtered supernatant. Data were analyzed for significance (*p* < 0.05) using a two-tailed Mann–Whitney test. RLU improvements were minimal for Delta (“ns” represents *p* > 0.05), while they were moderate in the case of D614G and BA4 (** represent *p* < 0.01), and maximum improvements (**** represent *p* < 0.0001) were observed for the Beta variant.

**Figure 3 viruses-16-01492-f003:**
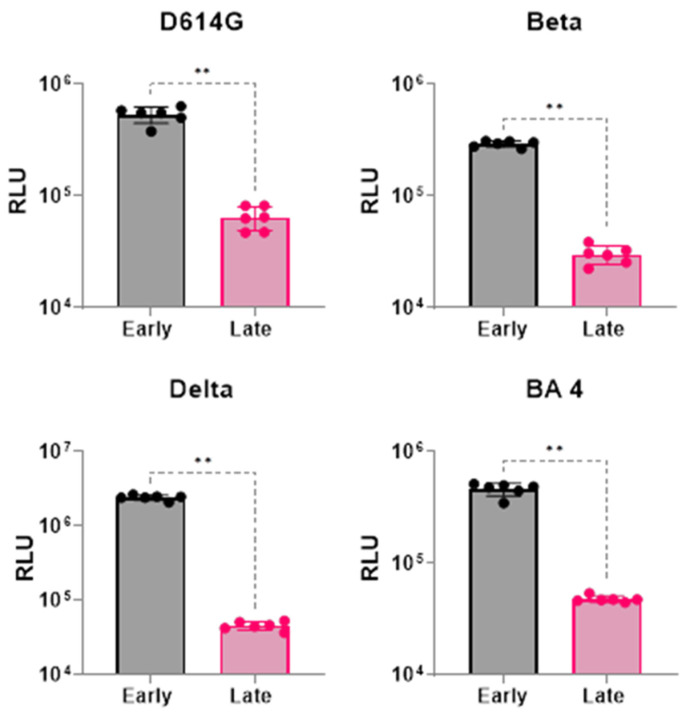
Host cell transduction timing after cell seeding (early vs. late transduction). Data were analyzed for significance (*p* < 0.05) using a two-tailed Mann–Whitney test. Compared to late transduction (16–18 h), early transduction (4–6 h) significantly (** represent *p* < 0.01) enhanced RLU numbers for all four variants studied. Data represents two different experiments with at least two replicates each.

**Figure 4 viruses-16-01492-f004:**
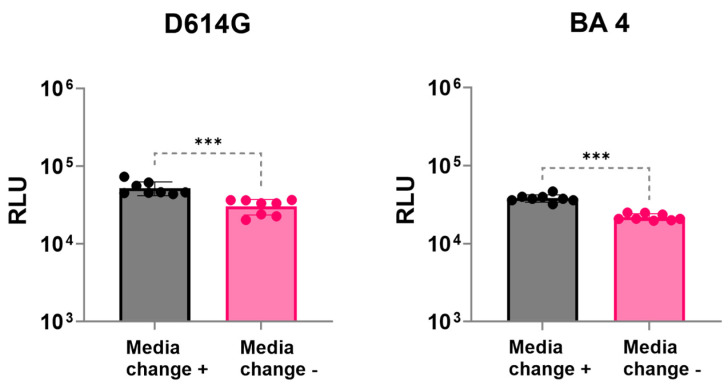
Replenishment of media. Changing the cell culture media (16–18 h) after pseudovirus transduction resulted in higher transduction of cells. Data were analyzed for significance (*p* < 0.05) using a two-tailed Mann–Whitney test. Media change significantly (*** represent *p* < 0.001) enhanced infectivity (transduction) numbers for both variants studied. The data represent two experiments with at least two replicates each.

**Figure 5 viruses-16-01492-f005:**
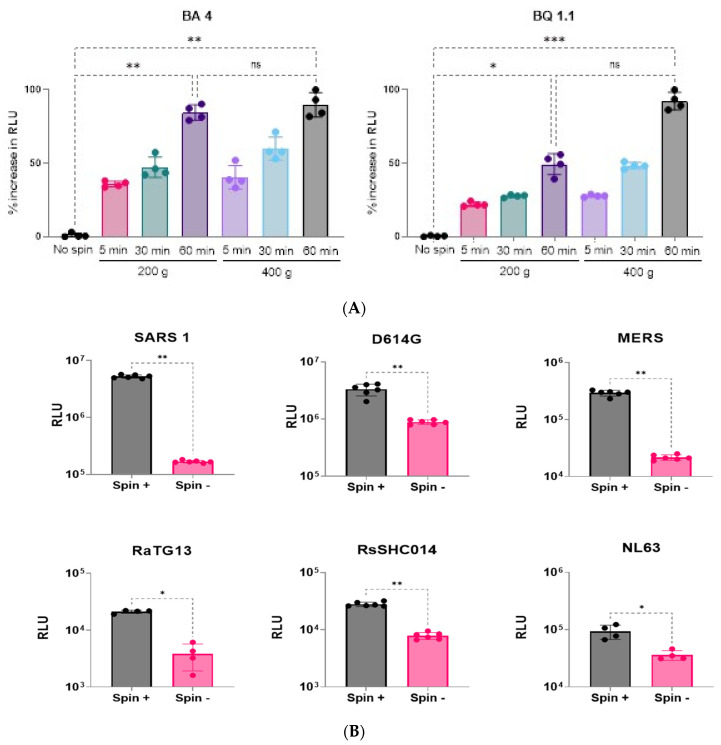
(**A**) Optimization of spinoculation speed and duration. The *X*-axis indicates a combination of centrifugation speeds (200 or 400× *g*) and duration (5, 30, or 60 min), while the *Y*-axis shows a % increase in RLU compared to no-spin controls. Data were analyzed for significance (*p* < 0.05) using a two-tailed Mann–Whitney test. For both variants studied, spinning for 60 min enhanced infectivity numbers significantly (* represent *p* < 0.05, ** represent *p* < 0.01, *** represent *p* < 0.001)); however, the spinoculation speed did not have a significant (“ns” represents *p* > 0.05) impact. (**B**) Spinoculation enhanced RLU numbers significantly (* represent *p* < 0.05, ** represent *p* < 0.01). The *X*-axis indicates either the presence or the absence of the spinoculation treatment (200× *g* for 60 min), while the *Y*-axis shows RLU. Data were analyzed for significance (*p* < 0.05) using a two-tailed Mann–Whitney test.

**Figure 6 viruses-16-01492-f006:**
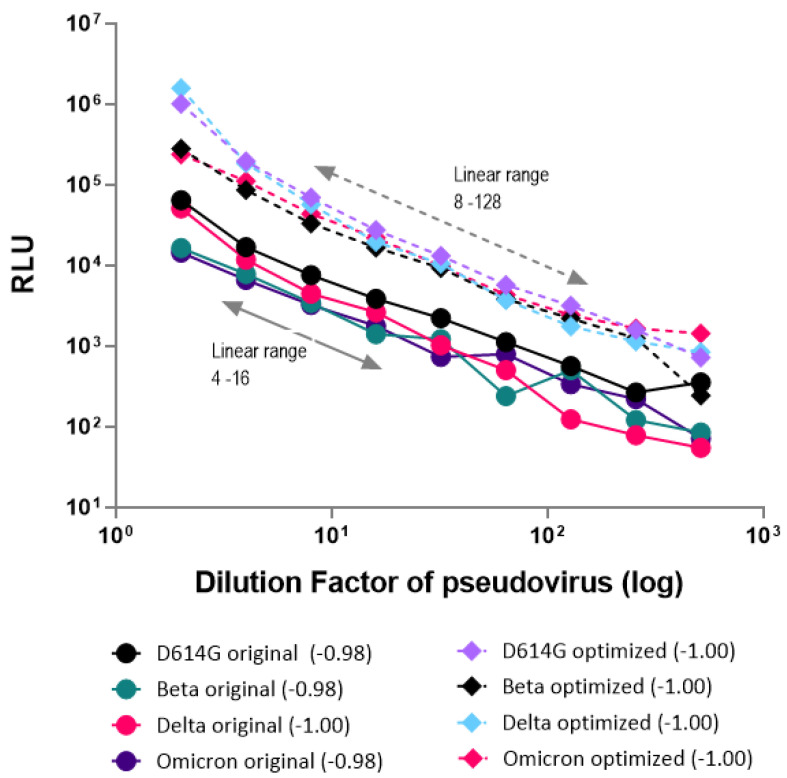
Comparison of the original protocol with the new protocol for linearity in the titration of pseudovirus particles. The *X*-axis shows the dilution factor of the pseudovirus used for transducing host cells, while the *Y*-axis shows luminescence (RLU). For all four variants studied, the new protocol enhanced RLU numbers by at least one log value. Both protocols indicated good linearity (regression R-value range in the parenthesis −0.98 to −1.0) for all four variants. However, the linearity improved with the new protocol (5 points spanning a dilution of 8–128), compared to the original protocol (3 points spanning a dilution of 4–16) due to increased dynamic range.

**Figure 7 viruses-16-01492-f007:**
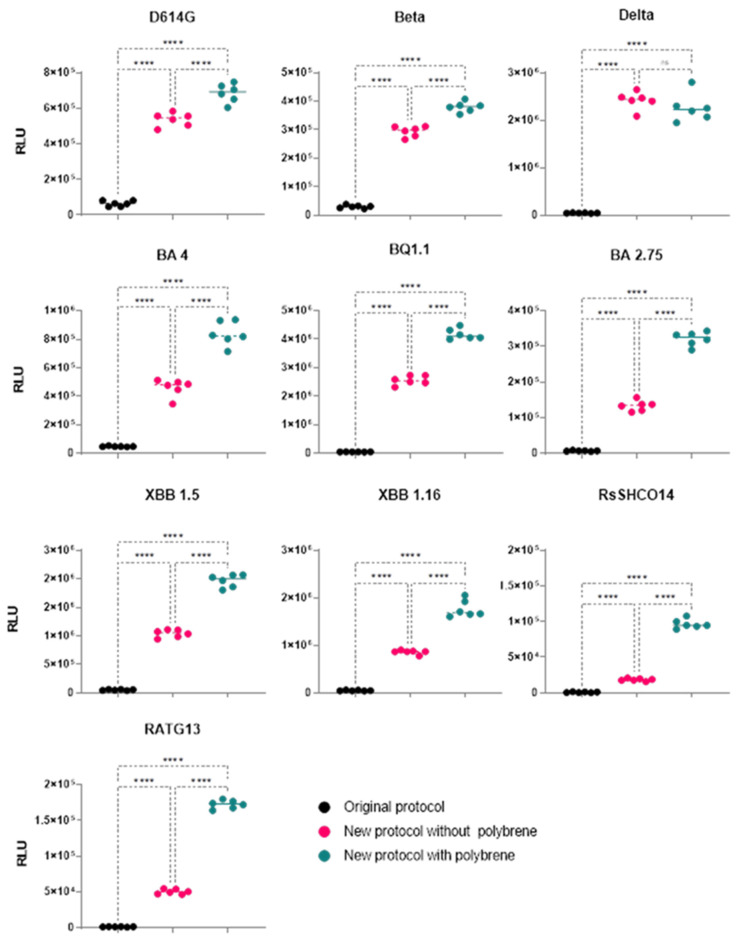
The effect of polybrene. RLU values (three different experiments with two replicates each) from the three protocols were analyzed for significance (*p* < 0.05) using one-way ANOVA and the Tukey test for multiple comparisons (“ns” represent *p* > 0.05, **** represent *p* < 0.0001). The new protocol improved transduction numbers by at least a log-fold for all ten variants studied Polybrene in combination with the new protocol resulted in similar (Delta) or better (rest of the variants) transduction numbers.

**Figure 8 viruses-16-01492-f008:**
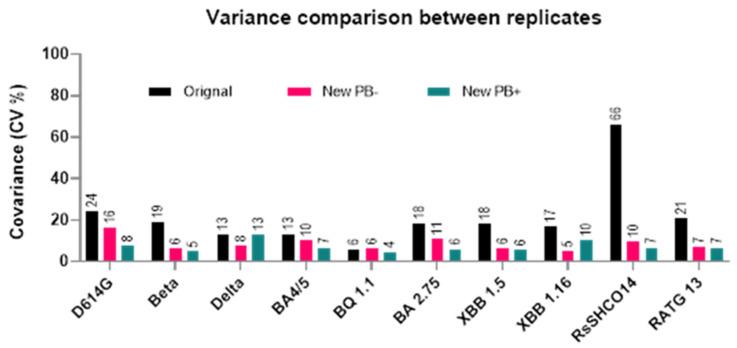
Coefficient of variation (CV%) analysis to assess variability within replicates in the original protocol, new protocol with polybrene, and new protocol without polybrene. CV% value is plotted on the *Y*-axis with each bar on the *X*-axis representing the value from 6 replicates of different spike variants. Variance observed using the original protocol ranged from 6 to 66%. With the new protocol (no polybrene), it ranged from 5 to 16%, and when polybrene was used in combination with the new protocol, the variance observed between the replicates was minimal at 4–13%.

**Table 1 viruses-16-01492-t001:** Key parameters for improved production and transduction efficiency of coronavirus pseudoviruses.

Parameter	Original Protocol	Optimized Conditions	Comments
Collection of cell-debris-free pseudovirus supernatant after transfection.	Filtration (0.45 μm) of supernatant before aliquoting and storing at −80 °C.	Centrifugation (8 min at 1200× *g*) of supernatant before aliquoting to store at −80 °C.	Applicable to all pseudoviruses.
Timing of transduction after cell seeding.	16–18 h after cell seeding.	4–6 h after cell seeding.	Applicable to all pseudoviruses and cell lines.
Media replenishment after viral adsorption.	No culture media replenishment until the end of 48 h incubation with pseudovirus.	Culture media replenished after 16–18 h of pseudovirus adsorption, and continued incubation for 48 h.	Applicable to all pseudoviruses and cell lines.
Spinoculation after the addition of viral particles to facilitate transduction.	No centrifugation (spinoculation) after transduction.	Centrifugation for 60 min at 200× *g* after transduction of host cells with pseudoviruses, and before 48 h incubation.	Applicable to all pseudoviruses and cell lines.
Addition of polybrene to the culture media during transduction.	Polybrene not used.	Polybrene addition at 4 μg/mL to the cell culture medium during transduction and neutralization.	Only applied to variant pseudoviruses that yielded low viral titers or that showed low RLUs in the viral controls during neutralization.

**Table 2 viruses-16-01492-t002:** Comparative infectivity (transduction efficiency) of coronavirus pseudoviruses generated using our new protocol.

S. No.	Pseudovirus Variant Name	Variant Classification	Avg TCID /mL of PV	Avg RLU /mL of PV
1	SARS 1	SAR-CoV-1	8.2 × 10^4^	3.7 × 10^8^
2	D614G	SARS-CoV-2	5.4 × 10^4^	3.3 × 10^8^
3	Beta	SARS-CoV-2	2.0 × 10^4^	1.4 × 10^7^
4	Delta	SARS-CoV-2	3.3 × 10^4^	2.0 × 10^7^
5	Omicron BA 4	SARS-CoV-2	4.1 × 10^4^	3.6 × 10^7^
6	Omicron BA 4.6	SARS-CoV-2	1.3 × 10^4^	1.1 × 10^7^
7	Omicron BA 2.75	SARS-CoV-2	7.2 × 10^3^	7.1 × 10^5^
8	Omicron BQ 1.1	SARS-CoV-2	2.9 × 10^4^	3.9 × 10^6^
9	Omicron XBB 1.5	SARS-CoV-2	1.1 × 10^5^	3.2 × 10^8^
10	Omicron XBB 1.16	SARS-CoV-2	3.6 × 10^4^	4.2 × 10^7^
11	EG.5	SARS-CoV-2	1.1 × 10^5^	1.1 × 10^8^
12	JN.1	SARS-CoV-2	9.8 × 10^4^	8.3 × 10^7^
13	MERS	Merbecovirus	1.0 × 10^4^	1.6 × 10^7^
14	NL-63	Alpha-Coronavirus	5.1 × 10^3^	8.3 × 10^5^
15	RaTG13	Bat Coronavirus	5.1 × 10^3^	1.8 × 10^6^
16	RsSHC014	Bat Coronavirus	4.3 × 10^3^	8.7 × 10^5^
17	WIV 1	Bat Coronavirus	1.2 × 10^5^	4.0 × 10^8^
18	WIV 16	Bat Coronavirus	5.8 × 10^4^	3.7 × 10^8^

## Data Availability

The data supporting this study’s findings are available from the corresponding author upon reasonable request.

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
