# Peer review of "Optimization of Cellular Transduction by the HIV-Based Pseudovirus Platform with Pan-Coronavirus Spike Proteins"

_viruses, 2024, doi:10.3390/v16091492_

Round 1

Reviewer 1 Report

Comments and Suggestions for Authors

The manuscript by Thimmiraju and colleagues describes a method to improve the production of coronavirus glycoprotein-pseudotyped HIV vectors and neutralization tests with the vectors. To this end, the standard procedure is modified at several points and the effect on the concentration and quantity of pseudovirions produced and on transduction efficiency is investigated. The manuscript contains a protocol for the optimal production of the vectors and the neutralization test.

The article is detailed, interesting and well written. The study is stimulating and useful for those working with HIV pseudovirions and for laboratories performing neutralization assays with pseudotyped viral constructs.

I have the following comments:

Page 3, 

Line 106: ... reporter plasmid ... The correct spelling for the plasmid is "pNL4-3.Luc.R-E-".

Line 107: "...spike protein expression plasmid, pcDNA3.1". Since the plasmid does not contain a protein gene, it is more likely to mean "... spike protein based on the expression plasmid pcDNA3.1".

Page 6, 

Line 41: "... dilution of heat-inactivated test sera samples (10 μL of sera and 90 μL of DMEM)". Was DMEM used with or without FBS? What type of 96-well culture plates were used for the dilutions and for the incubation of serum and pseudovirions?

Page 7:

Line 72ff: As I understand it, the Reed and Munch method was not used here to determine the NT50. The Reed and Munch method determines the concentration at which 50 % of the cultures are protected from infection. Instead, the method in the manuscript examined the concentration at which the RLU value in the cultures was reduced by 50 %. This is a common method and a reference is quoted in the text (Nie al. 2020), but not according to Reed and Munch.

Author Response

REVIEWER1:

The manuscript by Thimmiraju and colleagues describes a method to improve the production of coronavirus glycoprotein-pseudotyped HIV vectors and neutralization tests with the vectors. To this end, the standard procedure is modified at several points and the effect on the concentration and quantity of pseudovirions produced and on transduction efficiency is investigated. The manuscript contains a protocol for the optimal production of the vectors and the neutralization test.

The article is detailed, interesting and well written. The study is stimulating and useful for those working with HIV pseudovirions and for laboratories performing neutralization assays with pseudotyped viral constructs.

We thank you for your encouraging feedback.

I have the following comments:

Page 3, 

Line 106: ... reporter plasmid ... The correct spelling for the plasmid is "pNL4-3.Luc.R-E-".

We corrected the sentence.

Line 107: "...spike protein expression plasmid, pcDNA3.1". Since the plasmid does not contain a protein gene, it is more likely to mean "... spike protein based on the expression plasmid pcDNA3.1".

 We changed the sentence to convey the message correctly.

Page 6, 

Line 41: "... dilution of heat-inactivated test sera samples (10 μL of sera and 90 μL of DMEM)". Was DMEM used with or without FBS? What type of 96-well culture plates were used for the dilutions and for the incubation of serum and pseudovirions?

We incorporated all the further details as suggested by the reviewer.

Page 7:

Line 72ff: As I understand it, the Reed and Munch method was not used here to determine the NT50. The Reed and Munch method determines the concentration at which 50 % of the cultures are protected from infection. Instead, the method in the manuscript examined the concentration at which the RLU value in the cultures was reduced by 50 %. This is a common method, and a reference is quoted in the text (Nie al. 2020), but not according to Reed and Munch.

We appreciate the suggestion. We removed the wording (Reed Munich method) that referred to the calculation of TCID50 and reframed the sentence to reflect the IC50 method described in the quoted reference.

Reviewer 2 Report

Comments and Suggestions for Authors

In this study, Jeroen Pollet and colleagues described optimization of lentivirus production for HIV-1-based viruses that were pseudo typed with coronavirus spike proteins. The authors provide data on the various parameters of virus preparation, including time of collection, various types to supernatant clarification and different ways to transduce the target cells. The authors demonstrated that using optimized protocol and improved transduction protocol, which included spinoculation and polybrene addition results in significantly higher efficacy and better utility for these viruses to be used in neutralization assays. In general, this is a useful study, which can inform on the improvement of preparation of coronavirus spike-pseudo typed viruses that are needed to improve BSL-2 based coronavirus assays and testing. The drawback of the study is that it misses details on transfection. It lacks p24 analysis that is required to assess the efficiency of pseudo virus production. Surprisingly, the authors has not tried to concentration the viruses to improve the efficiency of transduction. Please see comments below.  

Major critics

1.       Table 1, centrifugation at 1200 g for 8 min is not sufficient to concentrate lentiviruses. You will need 35,000 x g for 8-12 hrs. Thus, the first line is incorrectly labeled as Method of concentrating.

2.       For lentivirus production, Lenti-x 293T cells are superior to the HEK-293T cells.

3.       How did you transfect the cells? Did you check the effect of the envelope on the viral titer collected?  What was the percent of confluence during transfection?  It is very unusual to have lower viral titers at 72 hrs compared to 48 hrs. This could be an artifact of overconfluent cells, for example.

4.       Please compare your viral titers in supernatant in the VSVg-pseudotyped HIV-1-Luc, which typically gives a higher titer.

5.       You did not concentrate the viruses, which is typically done at 35,000 x g for 8-12 hrs. Your 1200 g, 8 min centrifugation is not capable of precipitating viruses.

6.       Results, section 3.1. Please provide p24 values for the collected viruses. In our hands, 72 hrs time point gives better titer than 49 hrs. Showing RLUs is not informative.

7.       Results section 3.2. What was the percent of confluence? If you overseed the cells, then the “late” cells would be overgrown. Also, your naming “late” and “early” is counter intuitive. I would switch it around.

8.       Figure 4, there seems to be no biologically significant difference (i.e. 3-fold or more). Minor statistically significant changes are irrelevant.

9.       Results section 3.4.: spinoculation is typically done at 1000xg. Your lower g may reduce the efficiency of spinoculation.

10.   Discussion, lane 260: VSVg-based pseudovirus platform is the HIV-1 based. Unless you pseudo typed it with HIV-1 envelope. I am not sure what the authors are talking about here.

Minor critics

1.       Please indicate the corresponding author(s).

2.       Some of the references are presented in superscript (see Refs in lane 1 to lane 50). Please correct.

Comments on the Quality of English Language

1.       Lane 50, “for purpose[11],[12],[13].” Change to “for this purpose[11],[12],[13].”

Author Response

In this study, Jeroen Pollet and colleagues described optimization of lentivirus production for HIV-1-based viruses that were pseudo typed with coronavirus spike proteins. The authors provide data on the various parameters of virus preparation, including time of collection, various types of supernatant clarification and different ways to transduce the target cells. The authors demonstrated that using optimized protocol and improved transduction protocol, which included spinoculation and polybrene addition results in significantly higher efficacy and better utility for these viruses to be used in neutralization assays. In general, this is a useful study, which can inform on the improvement of preparation of coronavirus spike-pseudo typed viruses that are needed to improve BSL-2 based coronavirus assays and testing. The drawback of the study is that it misses details on transfection. It lacks p24 analysis that is required to assess the efficiency of pseudo virus production. Surprisingly, the authors has not tried to concentrate the viruses to improve the efficiency of transduction. Please see comments below.  

 Major critics

  1. Table 1, centrifugation at 1200 g for 8 min is not sufficient to concentrate lentiviruses. You will need 35,000 x g for 8-12 hrs. Thus, the first line is incorrectly labeled as Method of concentrating.

Thank you for this valuable feedback. We did not attempt to concentrate the pseudovirus, which would require ultra-centrifugation speeds of 35000+g, as pointed out by the reviewer. We did the centrifugation to clarify the pseudovirus supernatant from the cell debris [1] by applying a low centrifugation speed of 1200g. We revised our wording in the manuscript to make this clearer.

  1. For lentivirus production, Lenti-x 293T cells are superior to the HEK-293T cells.

Thank you for the suggestion. We agree Lenti-x 293T would have been an excellent choice. However, based on our institution's established protocols and available resources, we used HEK-293T cells.

  1. How did you transfect the cells? Did you check the effect of the envelope on the viral titer collected?  What was the percentage of confluence during transfection?  It is very unusual to have lower viral titers at 72 hrs. compared to 48 hrs. This could be an artifact of overconfluent cells, for example.

We seeded HEK-293T cells to ensure at least 60% confluency after 16-24 hours of incubation at 37°C, which is ideal for transfection. After confirming under the microscope that the cells were healthy and at the desired confluency, we proceeded with the transfection by adding the appropriate amounts of packaging, reporter, and spike plasmids. The cells were then incubated for 48 hours before collecting the pseudovirus. However, by 72 hours, the cells likely became over-confluent due to our initial seeding concentration. This may have contributed to poor cell health and lower viral titers observed at this time point compared to those at 48 hours.

  1. Please compare your viral titers in supernatant in the VSVg-pseudotyped HIV-1-Luc, which typically gives a higher titer.

As supported by the published literature, VSVg-pseudotyped viruses can also be used for pseudotyping coronaviruses and generally produce higher titers. However, the particle characteristics differ slightly. For ease of use and stability, we prefer the HIV-1-Luc system to mimic coronavirus particles. Consequently, we are focusing on optimizing the titers of the HIV-1 platform across various coronavirus spike variants.

  1. You did not concentrate the viruses, which is typically done at 35,000 x g for 8-12 hrs. Your 1200 g, 8 min centrifugation is not capable of precipitating viruses.

We did not attempt to concentrate the pseudovirus, which requires ultra-centrifugation speeds of 35000+g, as pointed out by the reviewer. We included a centrifugation step to clarify [2] the pseudovirus supernatant from the cell debris.

  1. Results, section 3.1. Please provide p24 values for the collected viruses. In our hands, 72 hrs time point gives better titer than 49 hrs. Showing RLUs is not informative.

We do agree that p24 values can be of value to quantify the pseudovirus production. However, our experience shows p24 values are not always well correlated with infectious titers. For a part of the dataset shown in the manuscript, we compared the TCID50/mL and RLU/mL data for pseudovirus supernatants collected @ 48h vs. 72h for four spike variants (as represented in the chart below). We have observed that the pseudovirus supernatant collected at 48h is more infective than the titer at 72h. 

  1. Results section 3.2. What was the percent of confluence? If you overseed the cells, then the “late” cells would be overgrown. Also, your naming “late” and “early” is counter intuitive. I would switch it around.

Host cell confluency was 40-60% before transduction. We labeled the time points as 'early' and 'late' based on the incubation duration from cell seeding to transduction (4-6 hours vs. 16-18 hours, respectively), as cells are more amenable for transduction at earlier time points.

  1. Figure 4, there seems to be no biologically significant difference (i.e. 3-fold or more). Minor statistically significant changes are irrelevant.

The purpose of media replenishment is to remove un-adsorbed pseudovirus, and cell debris to reduce cell toxicity.   Additionally, supplying fresh round of nutrients may contribute to better cell health and improved transduction efficiencies.

  1. Results section 3.4.: spinoculation is typically done at 1000xg. Your lower g may reduce the efficiency of spinoculation.

In the studies for this manuscript, we are optimizing multiple parameters to enhance transduction efficiency using a single cell line, HEK293T. Since 293T cells are typically loosely adhered to the surface, we chose a minimal spinoculation speed of 200xg and 400g  to minimize any possible cell loss.

  1. Discussion, lane 260: VSVg-based pseudovirus platform is the HIV-1 based. Unless you pseudo typed it with HIV-1 envelope. I am not sure what the authors are talking about here.

 We revised the sentence to improve clarity.

Minor critics

  1. Please indicate the corresponding author(s).

The corresponding author is indicated by *

  1. Some of the references are presented in superscript (see Refs in lane 1 to lane 50). Please correct.

References are now all indicated in the same style

[1] Jing Qu and Zhenhua Yang, “Protocol to Produce High-Titer Retrovirus for Transduction of Mouse Bone Marrow Cells,” STAR Protocols 2, no. 2 (April 7, 2021): 100459, https://doi.org/10.1016/j.xpro.2021.100459.

[ii] Caroline B. Plescia et al., “SARS-CoV-2 Viral Budding and Entry Can Be Modeled Using BSL-2 Level Virus-like Particles,” The Journal of Biological Chemistry 296 (November 27, 2020): 100103, https://doi.org/10.1074/jbc.RA120.016148.